# In Vitro and In Silico Pharmacological and Cosmeceutical Potential of Ten Essential Oils from Aromatic Medicinal Plants from the Mascarene Islands

**DOI:** 10.3390/molecules27248705

**Published:** 2022-12-08

**Authors:** Bibi Sharmeen Jugreet, Namrita Lall, Isa Anina Lambrechts, Anna-Mari Reid, Jacqueline Maphutha, Marizé Nel, Abdallah H. Hassan, Asaad Khalid, Ashraf N. Abdalla, Bao Le Van, Mohamad Fawzi Mahomoodally

**Affiliations:** 1Department of Health Sciences, Faculty of Medicine and Health Sciences, University of Mauritius, Réduit 80837, Mauritius; 2Department of Plant and Soil Sciences, University of Pretoria, Pretoria 0002, South Africa; 3School of Natural Resources, University of Missouri, Columbia, MO 65211, USA; 4College of Pharmacy, JSS Academy of Higher Education and Research, Mysuru 570015, India; 5Chemistry Department, College of Education, Salahaddin University, Erbil 44002, Iraq; 6Substance Abuse and Toxicology Research Center, Jazan University, Jazan 45142, Saudi Arabia; 7Medicinal and Aromatic Plants and Traditional Medicine Research Institute, National Center for Research, Khartoum P.O. Box 2404, Sudan; 8Department of Pharmacology and Toxicology, College of Pharmacy, Umm Al-Qura University, Makkah 21955, Saudi Arabia; 9Institute of Research and Development, Duy Tan University, Da Nang 550000, Vietnam; 10Faculty of Natural Sciences, Duy Tan University, Da Nang 550000, Vietnam; 11Center for Transdisciplinary Research, Department of Pharmacology, Saveetha Institute of Medical and Technical Science, Saveetha Dental College, Chennai 600077, India; 12Centre of Excellence for Pharmaceutical Sciences, North-West University, Private Bag X6001, Potchefstroom 2520, South Africa

**Keywords:** essential oils, antiacne, antimycobacterial, antiaging, antiproliferative

## Abstract

In this study, 10 essential oils (EOs), from nine plants (*Cinnamomum camphora*, *Curcuma longa*, *Citrus aurantium*, *Morinda citrifolia*, *Petroselinum crispum*, *Plectranthus amboinicus*, *Pittosporum senacia*, *Syzygium coriaceum*, and *Syzygium samarangense*) were assessed for their antimicrobial, antiaging and antiproliferative properties. While only *S. coriaceum*, *P. amboinicus* (MIC: 0.50 mg/mL) and *M. citrifolia* (MIC: 2 mg/mL) EOs showed activity against *Cutibacterium acnes*, all EOs except S. samarangense EO demonstrated activity against *Mycobacterium smegmatis* (MIC: 0.125–0.50 mg/mL). The EOs were either fungistatic or fungicidal against one or both tested Candida species with minimum inhibitory/fungicidal concentrations of 0.016–32 mg/mL. The EOs also inhibited one or both key enzymes involved in skin aging, elastase and collagenase (IC_50_: 89.22–459.2 µg/mL; 0.17–0.18 mg/mL, respectively). Turmerone, previously identified in the *C. longa* EO, showed the highest binding affinity with the enzymes (binding energy: −5.11 and −6.64 kcal/mol). Only *C. aurantium* leaf, *C. longa*, *P. amboinicus*, *P. senacia*, *S. coriaceum*, and *S. samarangense* EOs were cytotoxic to the human malignant melanoma cells, UCT-MEL1 (IC_50_: 88.91–277.25 µg/mL). All the EOs, except *M. citrifolia* EO, were also cytotoxic to the human keratinocytes non-tumorigenic cells, HaCat (IC_50_: 33.73–250.90 µg/mL). Altogether, some interesting therapeutic properties of the EOs of pharmacological/cosmeceutical interests were observed, which warrants further investigations.

## 1. Introduction

Natural products derived from plants are part of traditional medicine and represent therapeutic possibilities for treating a panoply of diseases. In recent years, their uses in the development of new drugs have shown much visibility due to their efficiency and limited adverse effects. Essential oils (EOs), which are aromatic and volatile liquids extracted from plants, are particularly indicated to possess a broad spectrum of curative properties, such as antimicrobial, antiviral, antimutagenic, anticancer, antioxidant, anti-inflammatory, immunomodulatory, and antiprotozoal properties [1]. In fact, they have been used for centuries in medicine, perfumery, and cosmetics, and have been added to foods as components of spices and herbs. Almost 3000 different EOs are known, and 300 are used commercially in the flavours and fragrances market [2].

Moreover, their molecular diversity, wide range of activity, structure–activity relationships, and capacity for targeting paradoxical responses triggered by different genes and pathways, have been significantly appraised [3]. A greater understanding of EOs’ chemistry and penetrative capabilities via biological membranes make them important treatment tools for the management of various neurological disorders. Essential oils in combination with vegetable oils are used in massages, with some being reported to cure one or more diseases and are used in para-medicinal practices [4,5].

Additionally, EOs have been shown to possess anticancer properties through various mechanisms, including cancer preventative mechanisms, as well as acting on the established tumor cell itself as well as interaction with the microenvironment [6]. Importantly, these activity mechanisms of EOs lead to cellular and metabolic responses, thus making them attractive in anticancer therapeutic strategies.

Furthermore, EOs, as complex mixtures of volatile compounds, can be regarded as a powerful tool to reduce bacterial resistance. An important characteristic of EOs and their components is their hydrophobicity, which enables them to partition with the lipids present in the cell membrane of bacteria and mitochondria, rendering them more permeable by disturbing the cell structures. This eventually results in the death of bacterial cells due to substantial leakage of critical molecules and ions from the bacterial cells. Some EO compounds have also been found to modulate drug resistance by targeting efflux mechanisms in several species of Gram-negative bacteria [7]. Additionally, EOs have been acknowledged to act as antifungal agents, and play an important role in blocking cell communication mechanisms, fungal biofilm formation, and mycotoxin production [8].

Indeed, botanical products that can prevent or reduce the signs of aging skin include products that offer photoprotection, decreased transepidermal water loss, increased skin elasticity, collagen formation, and decreased facial pigmentation, or offer antioxidant effects in the skin. Essential oils are no exception, as they have been found to counteract some of the signs of skin aging [9]. Besides, most of these oils also confer powerful antioxidant benefits, which means they have the power to scavenge free radicals to protect the skin from damage [10].

In this study, 10 EOs extracted from nine endemic and exotic medicinal plants from Mauritius, which have previously been found to possess multiple benefits in vitro and showed promising results in silico [11,12,13], were subjected to further investigations. Notably, they were explored for their antimicrobial, antiaging, and antiproliferative properties.

## 2. Results and Discussion

### 2.1. Antimicrobial

#### 2.1.1. Antimycobacterial

*Mycobacterium* species are responsible for several diseases, particularly in immunocompromised individuals. The spread of resistance to antimycobacterial drugs is a significant problem to public health and requires the search for a new and innovative alternative for the treatment of drug-resistant mycobacterial strains [14]. The hydrophobic structure of the cell wall is responsible for the innate antibiotic resistance of *Mycobacterium* species. It has been suggested that they became more hydrophobic by increasing the proportion of less polar lipids in their outer membrane. Importantly, such a change implies an enhanced capability for aerosol transmission, affecting their virulence and pathogenicity [15]. Since it is impermeable for commonly used antibiotics and can be attacked by substances with a high affinity for lipid-rich cell surfaces, one successful route to overcoming the hydrophobic barrier of the mycobacterial outer membrane is to use hydrophobic biologically active compounds, such as EOs [16]. Thus, in this study, the inhibitory effect of the EOs on *Mycobacterium smegmatis* was evaluated. Indeed, *M. smegmatis* is a useful research surrogate for pathogenic *Mycobacterium* species in a laboratory experimental setup [17], given that working with some strains of *Mycobacterium*, such as *M. tuberculosis*, poses some biosafety risks [18].

In the present study, with the only exception of *S. samarangense* EO, all EOs demonstrated antimycobacterial potential against *M. smegmatis* (MIC range: 0.125–0.50 mg/mL). In particular, *S. coriaceum* EO exhibited the most potent inhibition against *M. smegmatis* (MIC: 0.125 mg/mL), although it was still lower compared with the antibiotic ciprofloxacin (Table 1). Many EOs have also been reported to be effective against both tuberculous and non-tuberculous mycobacteria [14,16,19,20], and thus they can be considered as important antimycobacterial agents from natural products.

#### 2.1.2. Anti-Acne

Acne is a common chronic inflammatory skin disease in both adolescents and adults that mainly involves the epidermis and pilosebaceous units. The pathogenesis of acne is complicated and the colonization of *Cutibacterium acnes* is considered a crucial factor throughout the whole development of acne. *Cutibacterium acnes* promotes the abnormal proliferation and differentiation of keratinocytes and increases sebum production [21]. Furthermore, while a direct involvement of *C. acnes* seems certain, it may not be involved in the initiation of acne lesions, but instead would mediate later inflammatory events, causing deterioration of the lesions while stimulating the production of host antimicrobial peptides: small molecules with antimicrobial activity and immunomodulatory properties. Moreover, a lipase secreted by the bacterium is responsible for the hydrolysis of sebum and the subsequent release of free fatty acids, which results in an irritating and proinflammatory effect [22]. The main therapy for acne is topical and oral antibiotics, which have been used for decades against *C. acnes*. However, like many other bacteria, the increasing number of strains of drug-resistant *C. acnes* due to antibiotic abuse in acne treatment has aroused wide concern [23,24]; consequently, novel therapies are in high demand worldwide.

Medicinal plants, in particular, may present a unique source of new therapeutic options. Many studies have been dedicated to the documentation of traditional uses of medicinal plants for managing dermatological conditions, and these may represent a strong armor in drug discovery in the search for effective acne treatment [25,26]. Many EOs from medicinal plants have also been explored for their wide application in acne management, especially as topical preparations [27,28,29], and hence continue to be the focus of many anti-acne studies.

In the current investigation, only *P. amboinicus*, *M. citrifolia* and *S. coriaceum* EOs showed anti-acne activity. *P. amboinicus* and *S. coriaceum* EOs showed higher potential (MIC: 0.50 mg/mL) than *M. citrifolia* EO (2 mg/mL) (Table 1). This could be related to the presence of specific compounds effective against *C. acnes* present in them. It is well known that in acne, the reactive oxygen species (ROS) produced by neutrophils play a critical role in the irritation and destruction of the follicular wall and are responsible for the progression of acne [22,30]. In addition, *C. acnes* has been shown to survive for long periods of time in human tissues with low oxidative potential [31]. Therefore, EOs with antioxidant activity could have beneficial effects in acne management. In fact, in our recent studies, *P. amboinicus* and *S. coriaceum* EOs have demonstrated good antioxidant activity in various in vitro antioxidant assays performed [12,13]. As expected, the known antibacterial drug used as a positive control, notably, tetracycline demonstrated prominent inhibitory capacity (MIC: 7.8 × 10^−4^ mg/mL).

#### 2.1.3. Antifungal

Although *Candida* species are common commensals in the human microbiome, they can nevertheless trigger opportunistic fungal infections despite their non-pathogenic character when the immune system of the affected individuals is impaired or weakened. Moreover, the emergence of antifungal drug resistance in *Candida* species has led to increased morbidity and mortality in immunocompromised patients [32,33]. Thus, given the antifungal drug resistance patterns making treatment of candidiasis difficult, it is essential to search for new antifungals against these opportunistic human pathogens responsible for frequent nosocomial infections. In this context, plant-derived products including EOs have demonstrated promising results in vitro and in vivo against several *Candida* species and thus, could be considered useful in the development of novel anticandidal drugs [34,35].

In the present study, while *P. senacia* EO was inactive, the other EOs showed varied inhibitory potential against *C. albicans* (MIC range: 0.25–32 mg/mL). In particular, *M. citrifolia* EO was the most potent against *C. albicans* (0.25 mg/mL), followed by *P. amboinicus* (2 mg/mL), *P. crispum* and *C. aurantium* (leaf) (4 mg/mL) EOs. Among them, *M. citrifolia*, *P. amboinicus* and *C. aurantium* (leaf) EOs were found to be fungicidal at their MIC. Both *Syzygium* EOs showed an MIC of 8 mg/mL, although *S. coriaceum* EO was fungicidal while *S. samarangense* EO was only fungistatic at that concentration (Table 2).

Moreover, *C. tropicalis* was found to be more sensitive to the EOs. For instance, seven EOs, namely *C. aurantium* (leaf), *M. citrifolia*, *C. longa*, *P. amboinicus*, *P. crispum* and the two *Syzygium* Eos, demonstrated MICs ranging from 0.016 to 2 mg/mL and MFCs ranging from 0.0625 to 2 mg/mL. On the other hand, *C. camphora* and *C. aurantium* fruit peel EOs displayed an MIC of 8 mg/mL and a MFC of 8 and 16 mg/mL, respectively. Clearly, *C. aurantium* leaf EO was the most potent against *C. tropicalis* (MIC = 0.016 mg/mL and MFC = 0.0625 mg/mL), followed by *M. citrifolia* EO (MIC = 0.25 mg/mL, MFC = 1 mg/mL). *P. senacia* EO showed the weakest antifungal effect on *C. tropicalis* (MIC = 32 mg/mL) (Table 2). Interestingly, although the standard antifungals, nystatin and amphotericin B, were very effective against both *Candida* species, they were nevertheless fungistatic at their respective MIC values (Table 2).

In the present study, *M. citrifolia* EO was found to be effective against both *Candida* species, with an MIC of 0.25 mg/mL. This was in agreement with the study by Holanda et al. [36], which also reported the high efficacy of *M. citrifolia* fruit EO against *Candida* species, namely *C. albicans* and *C. utilise*, with MIC values of 39 and 78 μg/mL, respectively. Remarkably, the same major components, octanoic (38.7%) and hexanoic (20.0%) acids, were revealed in the EO, as was previously reported by Jugreet and Mahomoodally (2020) for *M. citrifolia* EO investigated in the present study [11], although they varied in their percentages (octanoic acid (78.9%); hexanoic acid (11.3%)). Furthermore, the antimicrobial potential of the oil was observed to decrease drastically after it was subjected to the esterification reaction, suggesting that the carboxyl group is responsible for the potent oil activity [36]. Hence, the good antifungal property obtained herein could be attributed to the richness of the *M. citrifolia* EO in short-chain fatty acids.

Furthermore, *C. aurantium* (leaf) EO was found to be the most potent against *C. tropicalis.* Interestingly, its major compound, sabinene, identified in our previous study [11], has also been found to be effective against several *Candida* species (MIC: 0.25 mg/mL), including *C. tropicalis* of the same ATCC strain used in this study and incubated for the same duration (24 h) [37]. However, the antifungal property of the EO was noted to be much more prominent in the present study compared to that of its major compound, sabinene, reported previously [37], which indicates that there could have been a synergistic interaction among the different components in the EO. Besides, *C. tropicalis* was reported to be more sensitive than *C. albicans* (same ATCC as the one used in this study), which was in agreement with the results obtained here, whereby *C. aurantium* (leaf) EO showed weaker inhibitory activity on *C. albicans.*

Similarly, some of the major components of the other EOs investigated here (ocimene, myrcene, carvacrol, pinene, limonene, and 1,8-cineole), were also evaluated in the study by İşcan [37] and were found to display anticandidal effects with MICs ranging from 0.12 to 4 mg/mL against *C. albicans* and *C. tropicalis.* The antifungal mechanisms of EOs and their constituents are normally explained by membrane damage or disruption of its integrity, increasing permeability, inhibition of ergosterol synthesis or binding to ergosterol on the membrane, and ROS production by acting on mitochondria [37]. Moreover, it has been suggested that EOs’ components accumulate in the lipophilic hydrocarbon molecules of the cell lipid bi-layer, thereby allowing the easier transfer of other components to the inner part of the cell [8].

### 2.2. Anti-Aging

The extracellular matrix (ECM) is the largest component of the dermis and provides the structural framework essential for the growth and elasticity of the skin. The ECM is composed of proteoglycans interwoven with macromolecules like collagen, elastin, and fibronectin, which are formed by the fibroblasts of the dermis. Collagen, the most abundant protein in the ECM, is responsible for the elasticity and strength of the skin and for maintaining its flexibility, while elastin confers the unique property of elastic recoil, which is vital for maintaining skin elasticity and resilience [38]. An increase in elastase activity has been found in several disorders, including psoriasis, dermatitis, inflammatory processes, and premature skin aging, which are closely associated with the formation of wrinkles [39]. Thus, the degradation of ECM is mainly due to the enhanced activity of proteolytic enzymes, such as collagenase and elastase.

The inhibition of these enzymatic activities by natural plant compounds is a promising approach to prevent skin aging and represents a pool of increasingly important ingredients in cosmetics and medications for the prevention of skin aging [40]. Of particular interest for anti-aging applications are the EOs, possessing multiple beneficial functions, such as the inhibition of aging-related enzymes and the capacity for scavenging free radicals [41,42,43].

In the present study, with the exception of four EOs (*C. aurantium* fruit peel, *C. camphora*, *M. citrifolia* and *P. amboinicus*), the remaining EOs were found to inhibit elastase (IC_50_: 141.81–588.80 µg/mL). On the other hand, all EOs showed interesting anti-collagenase activity, with IC_50_ values ranging from 0.17 to 1.54 mg/mL (Table 3). Remarkably, *C. aurantium* (leaf), *C. longa*, *P. crispum*, *P. senacia*, *S. coriaceum*, and *S. samarangense* EOs were found to inhibit both enzymes.

Of all the active EOs, *C. longa* EO displayed the most potent activity. In another study, the anti-aging effect of *C. longa* EO was also tested using a different strategy [44]. This was determined using ultraviolet B (UVB)-induced skin aging assays, whereby *C. longa* EO was seen to reduce cutaneous photoaging in a UVB-irradiated nude mouse model.

Indeed, EOs with anti-aging activities could be an interesting approach to combat skin aging and could be incorporated in topical creams [45]; the EOs under the present investigation provide good scope in this regard.

### 2.3. Molecular Docking

Molecular docking is one of the most applied virtual screening methods widely used in drug discovery [46]. This method helps to predict the intermolecular framework formed between a protein and a small molecule or a protein and protein and suggests the binding modes accountable for the inhibition of the protein [47]. Thus, molecular docking was selected as a suitable method to obtain useful information about the binding affinity of the tested components to the active site of the enzymes and therefore understand the interactions of the enzymes with the major oil components.

In the present study, the most abundant components of the EOs previously identified by GC-MS/GC-FID (in Appendix A) were docked with the enzymes’ active sites, with the results of the docking scores listed in Table 4. The lower the binding energy, the greater is the binding efficiency. As shown in Table 4, turmerone was revealed to be the most potent inhibitor and displayed the highest binding affinity with both collagenase and elastase (binding energy: −5.11 and −6.64 kcal/mol, respectively). The two-dimensional interactions of turmerone with the enzymes are depicted in Figure 1.

Carvacrol follows turmerone, with a binding energy of −4.55 kcal/mol with collagenase enzyme (Table 4). Myristicin and limonene were the compounds with the second- and third-highest binding affinity with elastase (−5.73 and −5.61 kcal/mol, respectively). Interestingly, turmerone was also reported to be a potent enzyme inhibitor in previous in silico studies [13], that could be related to the efficient binding interactions between the compound and these enzymes.

### 2.4. Cytotoxic/Antiproliferative Evaluation of EOs

Currently there is a need for novel non-toxic and selective agents to prevent and/or treat cancer and tumor malignancies. One method involves using synergistic molecules that can block multiple pathways. The use of active components derived from natural products is valuable in this regard [48]. Ancient practices using plants in the treatment of many diseases are increasingly being used in recent years. Medicinal plants are a source of compounds with biological activities as anticancer agents, and over 50% of the drugs used in the clinical treatment of cancer, such as taxol, camptothecin, vincristine, and vinblastine, are obtained from natural sources [49].

Key hallmarks of cancer include resisting cell death, sustained proliferative signalling, and escaping growth suppressors. Therefore, therapeutic strategies focused on inducing apoptosis and cellular arrest are of great significance. Remarkably, EOs have been shown to induce both the intrinsic (or mitochondria-dependent) and extrinsic (or death receptor-dependent) apoptosis pathways [50]. Interestingly, specific EO constituents have even been found to enhance the cytotoxic activity of chemotherapy drugs on various cell lines, thus increasing the therapeutic window; that is, showing the same effectiveness with reduced drug concentrations [51,52].

While the efficacy of natural products towards certain cancer types is encouraging, their toxicity towards normal healthy cells must remain low to obtain the highest level of efficacy and specificity towards cancer cells [53]. Thus, in the present study, the cytotoxic effects of EOs were investigated in vitro on both cancerous and non-cancerous cells using the MTT assay. In the current study, a human keratinocyte non-tumorigenic cell line (HaCat) and a human malignant melanoma cell line (UCT-MEL1) were used. The results are presented in Table 5.

All EOs except *M. citrifolia* EO, showed cytotoxicity towards HaCat cells (IC_50_: 34.17–250.90 µg/mL). The highest cytotoxic effect was exhibited by *C. aurantium* (leaf) and *S. coriaceum* EOs (IC_50_: 33.73 ± 7.06 µg/mL and 34.17 ± 5.32 µg/mL, respectively). This was followed by EOs such as *P. amboinicus*, *P. senacia*, *S. samarangense* and *C. longa* EOs respectively (IC_50_: 49.12–56.10 µg/mL). On the other hand, a lower cytotoxic potential was observed by *C. camphora*, *C. aurantium* (fruit peel) and *P. crispum* EOs on the HaCat cells (IC_50_: 104.50–250.90 µg/mL). The positive control Actinomycin D, a known anti-tumor drug, instead showed a significantly higher cytotoxic effect on HaCat compared to the EOs (IC_50_: 2.85 × 10^−2^ ± 8.49 × 10^−4^ µg/mL). Out of the tested EOs, only *C. longa*, *C. aurantium* (leaf), *P. senacia*, *P. amboinicus*, *S. coriaceum,* and *S. samarangense* EOs showed inhibitory activity against the UCT-MEL1 cells, with IC_50_ values ranging from 88.91 to 277.25 µg/mL, although the values were significantly higher than the positive control actinomycin D (IC_50_: 8.65 × 10^−3^ ± 1.13 × 10^−4^ µg/mL). On the other hand, *M. citrifolia* EO did not exert inhibitory effect on any of the cell lines at the highest tested concentration (400 µg/mL) (Table 5). Furthermore, while some of the tested EOs demonstrated cytotoxicity towards the tumorigenic cells, they were however not selective towards them as they also showed cytotoxicity towards the normal non-tumorigenic cells.

Indeed, as reported, the cytotoxic properties of EOs result from the complex interaction between the different classes of compounds, such as phenols, alcohols, esters, aldehydes, ketones, ethers, or hydrocarbons [54]. Additionally, in some cases, the cytotoxic activity is closely related to a few of the main oil components and it has also been found that some of these isolated compounds exert considerable cytotoxic properties when tested individually [55]. Nevertheless, the scarcer compounds could also be of importance, as the various molecules could synergistically act with the major compounds [56,57], while antagonistic interactions of the compounds have been recognized as well [58]. The wide variation in the chemical profile of EOs also means a great diversity in the mechanisms of action and molecular targets, whereby each compound can modulate or alter the effects of another compound. The main mechanisms that mediate the cytotoxic effects of EOs include the induction of cell death by activating apoptosis and/or necrosis processes, cell cycle arrest, and loss of function of vital organelles. Several of these effects are due to the lipophilic nature and low molecular weight of the main components, allowing them to cross cell membranes, alter membrane composition, and increase membrane fluidity, causing leakage of ions and cytoplasmic molecules [54]. Interestingly, all the three major types of EO constituents namely, phenols, aldehydes, and alcohols, have been reported to exert cytotoxic effects in this way [59]. Hence, the presence of numerous constituents that simultaneously interfere with multiple signaling pathways might be the key for overcoming the current limit of chemotherapeutic agents and particularly, the development of multidrug resistance [60].

Some of the EOs investigated in this study have also been previously subjected to cytotoxic studies using different cancer cell lines. In the study by Jacob and Toloue [48], the purified turmeric oil fractions containing α, β and ar-turmerones showed growth inhibitory activity against breast (SKBR-3), pancreatic (PANC-1), and prostate (PC-3) cancers, and reduced activity against a non-cancerous cell line (WI-38). Percent inhibition was suggested to be associated with the structural parameters of the turmerones. Furthermore, turmeric EO was found to have significant in vitro cytotoxic activity against Dalton’s lymphoma ascites cells (DLA) and Ehrlich ascites carcinoma (EAC) cancer cell lines (IC_50_ 8 μg and 18 μg, respectively). Oral administration of turmeric EO was found to significantly increase the life span (56.25%) of DLA-induced ascites tumour bearing mice, as well as significantly reducing the solid tumours [61]. Furthermore, *M. citrifolia* EO was reported to be cytotoxic to human colorectal carcinoma (HCT-116) and human breast carcinoma (MCF-7) cell lines, exhibiting IC_50_ values of 91.46 µg mL^−1^ and 78.15 µg mL^−1^, respectively [62]. The chemotherapeutic activity of the *P. amboinicus* EO on C57BL/6 mice injected with B16F-10 melanoma cell line was also revealed in a former study, whereby *P. amboinicus* EO (50 μg/dose) via i.p. was used as a treatment for 21 days [63]. Additionally, the daily topical treatment with *C. camphora* EO in the study of Moayedi et al. [64], was found to induce dramatic regression of pre-malignant skin tumors and a two-fold reduction in cutaneous squamous cell carcinoma in vivo. Interestingly, *C. camphora* EO was found to stimulate calcium signaling, resulting in calcineurin-dependent activation of nuclear factor for activated T cells; in cultured keratinocytes and *in vivo*, it induced transcriptional variations in immune-related genes, resulting in cytotoxic T cell-dependent tumor regression.

## 3. Materials and Methods

### 3.1. Plant Materials

Nine plants, of which two are endemic species; namely, *Syzygium coriaceum* J. Bosser & J. Gueho and *Pittosporum senacia* Putterl. subsp. *Senacia*, and seven exotic species; namely, *Cinnamomum camphora* (L.) Nees & Eberm, *Citrus aurantium* L., *Curcuma longa* L., *Morinda citrifolia* L., *Petroselinum crispum* (Mill.) Fuss, *Plectranthus amboinicus* (Lour.) Sprengel, and *Syzygium samarangense* (Blume) Merr. & L. M. Perry, were used based on their importance in traditional medicine, as documented in a previous publication [11]. The plant specimens were authenticated by a local botanist at the Mauritius Sugarcane Industry Research Institute (MSIRI) Herbarium, at Réduit, whereby an identification code was assigned for each of them.

The leaves of *Syzygium coriaceum* (MAU 0027510) and *C. camphora* (MAU 0027508), were collected from Monvert nature park (20°20′35.2″ S, 57°31′21.2″ E) in the month of October 2018, while *C. aurantium* (leaves or fruit peel; MAU 0027511), *M. citrifolia* (fruits; MAU 0027506), *P. amboinicus* (leaves; MAU 0027507) and *P. senacia* (fruits; MAU 0027512) was obtained from the university farm, Réduit (20°13′39″ S, 57°29′33″ E) in the month of May to September 2018. The leaves of *S. samarangense* (MAU 0027509) was collected in March 2018 from Mont-fertile, a southern region in Mauritius (20°24′31.0″ S, 57°36′49.1″ E). Rhizomes of *C. longa* (MAU 0027514) were harvested from Plaine-Magnien, a southern region in Mauritius (latitude 20°25′46.81″ S, longitude 57°40′10.85″ E) in May 2018. Lastly, *P. crispum* (aerial parts; MAU 0027505) was purchased in July 2018, from the local market.

### 3.2. Extraction of EOs

Fresh plant material, cut into small pieces, was subjected to the process of hydrodistillation using a Clevenger-type apparatus for 3 h. The EO distillates, once yielded (% yield in Appendix A), were dried over anhydrous magnesium sulfate, filtered, and then stored in dark vials at −4 °C until further analysis [65].

### 3.3. Antimicrobial Assay

#### 3.3.1. Anti-Mycobacterium

The minimum inhibitory concentration (MIC) values of all the EOs were determined according to the method used by Lall et al. [66]. The EOs were dissolved in 20% DMSO and sterile Middlebrook 7H9 media, and two-fold dilution was performed to produce final test concentrations ranging from 31.25 to 1000 µg/mL. The bacterial suspension was adjusted to 0.5 McFarland standard (1.5 × 10^8^ colony-forming units/mL [CFUs/mL]). The inoculum of *M. smegmatis* (ATCC^®^ (American Type Culture Collection, Rockville, MD, USA) MC^2^ 155) was further diluted 50-fold to obtain the final test concentration of (1.5 × 10^6^ CFUs/mL). After the addition of the bacterial inoculum (100 µL), the final assay volume in each well was 200 µL. Ciprofloxacin (Sigma-Aldrich, Saint Louis, MO, USA) (0.078 to 10 µg/mL) was used as standard drug. The plates were left to incubate at 37 °C for 24 h, followed by the addition of the viability indicator PrestoBlue^®^ (Invitrogen Corporation, San Diego, CA, USA) (20 µL) to each well, and after a further 2 h incubation, the colour change was observed. The MIC value was defined as the lowest concentration at which no colour change from blue to pink could be observed.

#### 3.3.2. Anti-Acne

The in vitro microdilution method as described by Kamatou [67] was followed to determine the MIC of the EOs against *Cutibacterium acnes* (ATCC^®^ 6919). The EOs were prepared in 100% acetone, with a final starting concentration of 2000 µg/mL. Tetracycline (Sigma-Aldrich, Saint Louis, MO, USA) served as the positive control, with a final starting concentration of 50 µg/mL. The sample dilutions were prepared in a 96-well flat-bottom microtiter plate. Brain–heart infusion broth (BHI; 100 µL) was added to all the wells, followed by 100 µL of the EOs in the first wells in triplicate. The EOs were diluted through a series of two-fold dilutions. The bacterial suspension was prepared from 72 h-old bacterial cultures grown on BHI agar at 37 °C under anaerobic conditions. The inoculated BHI broth was diluted to 6 × 10^6^ CFU/mL and added to all the wells (100 µL) except the broth control wells. After 72 h incubation under anaerobic conditions, 20 µL of PrestoBlue^®^ reagent was added to all the wells and incubated at 37 °C for 60 min. The MIC value was determined by observing the colour change.

#### 3.3.3. Antifungal

##### Microdilution Broth Susceptibility Assay

The MIC of the EOs was determined as previously described by Seebaluck-Sandoram et al. [68], with minor modifications. Each EO (100 µL) was serially diluted two-fold, in triplicate, with Mueller–Hinton broth (MHB) in 96-well microtitre plates. Fresh inoculums of *C. albicans* (ATCC 10231) and *C. tropicalis* (ATCC 750) were then prepared and adjusted to 0.5 Mc Farland standard, which were further diluted at a ratio of 1:100 with fresh broth in order to yield starting inoculums of approximately 10^6^ CFU/mL. Next, 100 µL of fungal culture was added to each well of the plates. Nystatin and amphotericin B (from Sigma-Aldrich Co., Steinheim, Germany) (100 µg/mL) were used as standard antifungal drugs. After 24 h incubation at 37 °C, 40 μL of iodonitrotetrazolium chloride (0.2 mg/mL) was added to each well and the plates were incubated for another 20 min. The well containing the lowest concentration in which no pinkish-red coloration was observed was regarded to be the MIC.

##### Minimum Fungicidal Concentration

The minimum fungicidal concentration (MFC) of the EOs was determined according to Aumeeruddy-Elalfi et al. [65]. Briefly, 10 µL of broth from the uncoloured wells (where no growth was observed in the earlier MIC assay), corresponding to the MIC value MIC × 2 (one dilution higher than MIC), and MIC × 4 (one dilution higher than MIC × 2), were inoculated on Sabouraud dextrose agar (SDA) and incubated at 37 °C for 24 h. The MFC was defined as the lowest recorded EO concentration of the MIC wells in which fungi failed to grow on the SDA. Alternatively, if growth was observed following inoculation on SDA, the concentration of the corresponding well used for inoculation (MIC, MIC × 2, and MIC × 4) was referred to as the fungistatic (FS) concentration. Both negative and positive controls were included for comparison.

### 3.4. Antiaging Assay

#### 3.4.1. Anti-Elastase

The method used to determine the anti-elastase potential of the EOs was that described by Lall et al. [69], with modifications. All reagents used were purchased from Sigma-Aldrich (Johannesburg, South Africa). Each EO and ursolic acid (positive control) were serially diluted in DMSO to obtain a final concentration range of 31.25–1000 µg/mL and 0.94–60 µg/mL, respectively. In a 96-well plate, 155 µL of potassium phosphate buffer (pH 8) was added, whereafter, 5 µL of each dilution was added (in triplicate) to the respective wells. Afterwards, 20 µL of 4.942 mU porcine pancreatic elastase enzyme was added and incubated at 37 °C for 5 min. Following incubation, the reaction was initiated by adding 20 µL of 4.4 mM *N*-succinyl-Ala-Ala-Ala-*ρ*-nitroanilide substrate. The absorbance values were measured using a BIO-TEK Power-Wave XS plate reader (Analytical and Diagnostic Products CC, Roodepoort, South Africa) at a wavelength of OD_405 nm_ for 15 min. The percentage inhibition was calculated using the following Equation (1) and GraphPad Prism 4 was used to determine 50% inhibitory concentration (IC_50_) for each sample:(1)$$\%\;\mathrm{Inhibition}=100-(\frac{\mathrm{Absorbance}\;\mathrm{sample}\;\mathrm{at}\;15\;\min-\mathrm{absorbance}\;\mathrm{at}\;0\;\min}{\mathrm{Absorbance}\;\mathrm{control}\;\mathrm{at}\;15\;\min-\mathrm{absorbance}\;\mathrm{at}\;0\;\min})\;\times \;100$$

#### 3.4.2. Anti-Collagenase

The anti-collagenase assay was performed as described previously by Aumeeruddy-elalfi et al. [43], with slight modifications using the EnzCheck^®^ Gelatinase/Collagenase Assay Kit (Molecular Probes Inc., Eugene, OR, USA). A 1× reaction buffer was prepared from 10× buffer provided in the kit. The enzyme collagenase from *Clostridium histolyticum* (Type IV) obtained in the kit was dissolved in the 1× reaction buffer for use at a concentration of 0.2 U/mL. Dye-quenched (DQ) gelatin from pigskin and fluorescein conjugate, used as the substrate, was diluted to a final concentration of 150 µg/mL. To constitute the reaction mixture, 80 µL of each EO dissolved in a 1× reaction buffer at different concentrations was distributed in Nunc 96-well microtitre plates. An amount of 100 µL of collagenase enzyme were added to the wells and allowed to incubate for 15 min at 37 °C. Following incubation, 20 µL of DQ gelatin was added to the reaction mixture. Fluorescence was read with parameters set for an excitation wavelength at 485 nm and emission at 515 nm. 1,10-phenanthroline was used as the positive control. Blank wells were prepared in the microplate, consisting of 100 µL 1× reaction buffer and 100 µL collagenase enzyme. The inhibition percentage of the collagenase enzyme was calculated using Equation (2) below [70]:Collagenase inhibition (%)  = {[(A−B)–(C−D)]/(A−B)} × 100(2)
where A is the fluorescent intensity without the test sample (control), B is the fluorescent intensity without the test sample and enzyme (blank of A), C is the fluorescent intensity with the test sample, and D is the fluorescent intensity with the test sample without enzyme. The anti-collagenase activities of all EOs were finally expressed as an IC_50_ value using GraphPad Prism 7 software.

### 3.5. Molecular Docking

The chemical structures of the most abundant compound in each EO (limonene, sabinene, 1,8-cineole, octanoic acid, carvacrol, myristicin, myrcene, (E)-β-ocimene β-pinene and turmerone) previously identified by GC-MS/GC-FID [11] (Appendix A), were downloaded as a mol file from Chemspider and ZINC database [71]. The AM1 method was applied using Gaussian09 software to optimize the structures [72]. The optimized structures were then each docked with the active sites of the enzymes elastase and collagenase.

The enzymes’ crystal structures were downloaded as a pdb format from the database, Protein Databank RCSB PDB. The pdb codes of the enzymes were 2Y6I for the collagenase and 1BRU for the structure of the porcine pancreatic elastase enzyme. Preparation of the protein structures for docking calculations was done as previously described [12]. All possible conformations were docked at the active site of the enzyme via the Lamarckian genetic algorithm of the AutoDock software, with 250 runs for each inhibitor. Visualisation and the docking results’ analysis were accomplished using the Discovery studio 5.0 visualizer.

### 3.6. Cell Culture

The human keratinocyte (HaCat) and human malignant melanoma (UCT-MEL-1) cells were maintained in T75 tissue-culture flasks containing Dulbecco’s Modified Eagles Medium (DMEM) supplemented with 1% antibiotics (100 U/mL penicillin and 100 µg/mL streptomycin), 250 µg/L fungizone, and 10% heat-inactivated and gamma-irradiated fetal bovine serum (FBS). The cells were grown in an incubator set to 37 °C and 5% CO_2_. The cells were passaged using 0.25% trypsin-EDTA after an 80% confluent monolayer had formed.

#### In Vitro Antiproliferative Activity

The in vitro antiproliferative activity of the EO samples was determined using the PrestoBlue viability assay, as described by Lall et al. [66], on human keratinocyte (HaCat) and human malignant melanoma (UCT-MEL-1) cell lines, donated by Dr Lester Davids from the University of Cape Town, Department of Human Biology (South Africa). The cells were seeded in 96-well microtitre plates at concentrations of 1 × 10^6^ cells/mL (UCT-MEL-1) and 4 × 10^4^ cells/mL (HaCat), and incubated for 24 h at 37 °C and 5% CO_2_ to allow for cell attachment. A stock solution of the EOs (40 mg/mL in DMSO) and the positive control, actinomycin D (Sigma-Aldrich, Saint Louis, MO, USA) (1 mg/mL in distilled water), was prepared, followed by serial dilutions to obtain final concentrations ranging from 12.5–400 µg/mL for the EOs, and 0.05–3.9 × 10^−4^ µg/mL for actinomycin D in the 96-well plates. The plates were incubated for 72 h, thereafter 20 µL PrestoBlue was added to each of the wells. After 2 h incubation, the fluorescence was measured at an excitation wavelength of 560 nm and an emission wavelength of 590 nm, using the VICTORNivo Multimode plate reader (Perkin Elmer, Midrand, South Africa). The samples were tested in triplicate to calculate percentage of cell viability using Equation (3) below and the IC_50_ values were determined using GraphPad Prism 9 software.
(3)$$\%\;\mathrm{Cell}\;\mathrm{viability}=\frac{\mathrm{Fluor}.\mathrm{sample}-\mathrm{Fluor}.0\%}{\mathrm{Fluor}.\mathrm{vehicle}\;\mathrm{control}-\mathrm{Fluor}.0\%}$$
where Fluor._sample_ is the fluorescence of (PrestoBlue + sample or the positive control) and Fluor_.0%_ is the fluorescence of (PrestoBlue + media), and Fluor._vehicle control_ is the fluorescence of (PrestoBlue + DMSO).

## 4. Conclusions

This study emphasized the antimicrobial, antiaging, and antiproliferative effects of plant-derived EOs. Interestingly, the EOs showed fungistatic and fungicidal effects on both tested *Candida* species. Additionally, some of the EOs were observed to also possess antimycobacterial and anti-acne properties against *M*. *smegmatis* and *C. acnes*, respectively.

Indeed, natural anti-aging skincare products have been in great demand in recent years, owing to their claimed effectiveness in delaying skin aging. Remarkably, the findings presented in the current study demonstrated that EOs could inhibit one or both key enzymes involved in skin aging, and thus could be regarded as attractive anti-aging ingredients. Additionally, molecular docking revealed the compound turmerone from *C. longa* EO to be the most potent inhibitor of the enzymes, which was also supported by the results from the in vitro anti-aging assays.

Furthermore, the in vitro antiproliferative properties of the EOs were tested using a human keratinocyte non-tumorigenic (HaCat) cell line and human malignant melanoma (UCT-MEL1) cell line. While most EOs showed a cytotoxic effect on the UCT-MEL1 melanoma cell line, they were not selective against the cancer cell and were quite cytotoxic to the HaCat cells. As EOs are complex mixtures of compounds, it would be interesting to test the cytotoxicity of only selected major components from the active EOs, as different components may exert antagonistic effects altogether, and thus decrease their overall effect on the cancer cells.

Taken together, this study revealed some interesting pharmacological and cosmeceutical attributes of the tested EOs that could be further exploited in in vivo assays to assess their mechanisms of action and unveil compounds of interest.

## Figures and Tables

**Figure 1 molecules-27-08705-f001:**
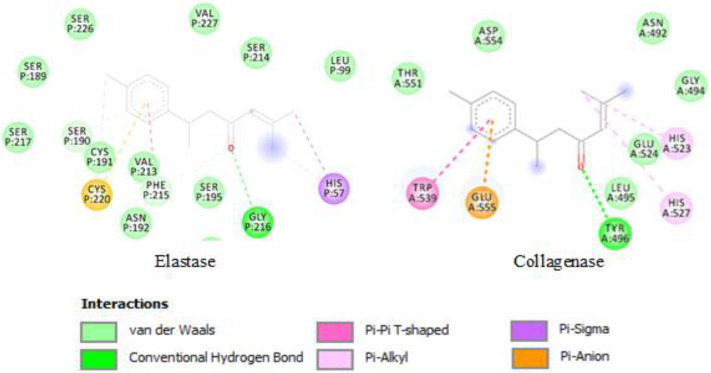
Shows intramolecular interactions of turmerone displaying the highest binding affinity with the studied enzymes.

**Table 1 molecules-27-08705-t001:** Antimycobacterial and anti-acne activities of EOs.

MIC (mg/mL)												
Bacteria Tested	Essential Oils										Antibiotics	
	CAF	CAL	CC	CL	MC	PA	PC	PS	SC	SS	CIP	TRC
*M. smegmatis* (ATCC MC^2^ 155)	0.25	0.25	0.50	0.25	0.25	0.25	0.25	0.50	0.125	NI_1_ ^b^	3.13 × 10^−4^	-
*C. acnes* (ATCC 6919)	NI_2_ ^c^	NI_2_ ^c^	NI_2_ ^c^	NI_2_ ^c^	2	0.50	NI_2_ ^c^	NI_2_ ^c^	0.50	NI_2_ ^c^	-	7.8 × 10^−4^

CC: *Cinnamomum camphora*; CAL: *Citrus aurantium* (leaf); CAF: *Citrus aurantium* (fruit peel), CL: *Curcuma longa*, MC: *Morinda citrifolia*, PC: *Petroselinum crispum*, PS: *Pittosporum senacia*, PA: *Plectranthus amboinicus*, SC: *Syzygium coriaceum*, SS: *Syzygium samarangense*; MIC: Minimum inhibitory concentration; ^b^ No inhibition at the highest concentration tested (1 mg/mL), ^c^ No inhibition at the highest concentration tested (2 mg/mL), CIP: ciprofloxacin, TRC: tetracycline.

**Table 2 molecules-27-08705-t002:** Antifungal activities of EOs against *Candida* spp.

MIC/MFC (mg/mL)												
Fungi Tested					Essential Oils						Antifungals	
	CAF	CAL	CC	CL	MC	PA	PC	PS	SC	SS	Nystatin	Amphotericin B
*C. albicans* (ATCC 10,231)	(32)	(4)	(16)	(8)	(0.25)	(2)	(4)	-	(8)	(8)	(1.56 × 10^−3^)	(3.13 × 10^−3^)
	FS	FC	FS	FS	FC	FC	FS		FC	FS	FS	FS
	[ND]		[32]	[16]			[16]			(16)	[3.13 × 10^−3^]	[ND]
*C. tropicalis* (ATCC 750)	(8)	(0.016)	(8)	(1)	(0.25)	(2)	(2)	(32)	(1)	(1)	(1.56 × 10^−3^)	(1.25 × 10^−2^)
	FS	FS	FC	FC	FS	FS	FC	FS	FS	FC	FS	FS
	[16]	[0.0625]			[1]	[4]		[ND]	[2]		[ND]	[ND]

MIC: minimum inhibitory concentration, MFC: minimum fungicidal concentration, CC: *Cinnamomum camphora*; CAL: *Citrus aurantium* (leaf); CAF: *Citrus aurantium* (fruit peel), CL: *Curcuma longa*, MC: *Morinda citrifolia*, PC: *Petroselinum crispum*, PS: *Pittosporum senacia*, PA: *Plectranthus amboinicus*, SC: *Syzygium coriaceum*, SS: *Syzygium samarangense*, - not active; ( ): MIC, FS: fungistatic at MIC, FC: fungicidal at MIC, [ ] new MFC in case not fungicidal at MIC, [ND]: not determined if MFC > MIC × 4.

**Table 3 molecules-27-08705-t003:** Anti-elastase and anti-collagenase activity of studied EOs.

EO	Elastase Inhibition	Collagenase Inhibition
	IC_50_ (µg/mL)	IC_50_ (mg/mL)
CAF	NI_1000_	1.46 ± 0.19
CAL	275.95 ± 13.86	1.54 ± 0.40
CC	NI_1000_	1.34 ± 0.09
CL	89.22 ± 23.72	0.17 ± 0.01
MC	NI_1000_	0.62 ± 0.04
PA	NI_1000_	0.33 ± 0.02
PC	354.65 ± 21.43	0.37 ± 0.02
PS	233.47 ± 21.45	0.77 ± 0.17
SC	767.2 ± 27.99	0.84 ± 0.13
SS	459.2 ± 21.24	0.18 ± 0.04
**Positive control**		
Ursolic acid	10.10 ± 15.27	-
1,1 Phenanthroline	-	4.20 × 10^−3^ ± 0.00

CAL: *Citrus aurantium* leaf, CAF: *Citrus aurantium* fruit (peel), CC: *Cinnamomum camphora*; CL: *Curcuma longa*, MC: *Morinda citrifolia*, PA: *Plectranthus amboinicus*, PC: *Petroselinum crispum*; PS: *Pittosporum senacia*; SC: *Syzygium coriaceum*; SS: *Syzygium samarangense*; IC50: half-maximal inhibitory concentration.

**Table 4 molecules-27-08705-t004:** Docking scores of EOs compounds with target enzymes.

EOs	Major Compounds	Collagenase	PP Elastase
CAF	Limonene	−3.87 ^a^ (1.5 mM) ^b^	−5.61 (77.7 µM)
CAL	Sabinene	−3.87 (1.5 mM)	−5.36 (118.8 µM)
CC	1,8-Cineole	−4.07 (1.0 mM)	−5.30 (130.8 µM)
CL	Turmerone	−5.11 (179.0 µM)	−6.64 (13.6 µM)
MC	Octanoic acid	−3.67 (2.1 mM)	−4.91 (249.7 µM)
PA	Carvacrol	−4.55 (459.4 µM)	−5.45 (100.9 µM)
PC	Myristicin	−3.56 (2.4 mM)	−5.73 (63.0 µM)
PS	Myrcene	−3.32 (3.7 mM)	−4.68 (374.3 µM)
SC	(E)-β-Ocimene	−3.29 (3.9 mM)	−4.63 (401.0 µM)
SS	β-Pinene	−4.13 (938.4 µM)	−5.47 (97.0 µM)

CAL: *Citrus aurantium* leaf, CAF: *Citrus aurantium* fruit (peel), CC: *Cinnamomum camphora*; CL: *Curcuma longa*, MC: *Morinda citrifolia*, PA: *Plectranthus amboinicus*, PC: *Petroselinum crispum*; PS: *Pittosporum senacia*; SC: *Syzygium coriaceum*; SS: *Syzygium samarangense*. ^a^ binding free energy in kcal/mol; ^b^ calculated inhibition constant.

**Table 5 molecules-27-08705-t005:** Cytotoxic effects of EOs on HaCat and UCT-MEL1 cell lines.

EOs	IC_50_ ± SD (µg/mL)	
	HaCat	UCT-MEL1
CAF	182.70 ± 3.54	NI_400_
CAL	33.73 ± 7.06	277.25 ± 1.48
CC	250.90 ± 0.57	NI_400_
CL	56.1 ± 1.90	88.91 ± 5.83
MC	NI_400_	NI_400_
PA	49.12 ± 2.58	189.50 ± 1.41
PC	104.50 ± 4.24	NI_400_
PS	50.33 ± 1.43	95.52 ± 0.77
SC	34.17 ± 5.32	95.37 ± 4.34
SS	54.70 ± 3.59	94.09 ± 1.85
ActinomycinD	2.85 × 10^−2^ ± 8.49 × 10^−4^	8.65 × 10^−3^ ± 1.13 × 10^−4^

Cytotoxicity is expressed as the concentration of the EOs inhibiting cell growth by 50% (IC_50_); NI_400_: no inhibition at the highest concentration tested of 400 µg/mL.

## Data Availability

Not applicable.

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
