# Peer review of "In Vitro and In Silico Pharmacological and Cosmeceutical Potential of Ten Essential Oils from Aromatic Medicinal Plants from the Mascarene Islands"

_molecules, 2022, doi:10.3390/molecules27248705_

Round 1

Reviewer 1 Report

Dear authors.

The work submitted for evaluation is interesting. It deals with the evaluation of the activity of oxygenated oils from plants growing in a specific local environment, largely isolated from external influences.

The work presented is methodically well prepared and the analysis presented in a multidirectional manner allows the evaluation of the material studied. The cell culture studies are well done and the researchers do not look for effects where there are none. I believe that the oils studied can be successfully applied in natural cosmetics with protective and regenerative effects. 

I would kindly ask you to supplement the information on the origin of the cell lines - the collection from which they were purchased and information on the software in which the calculations of the presented results were performed. 

Author Response

The work submitted for evaluation is interesting. It deals with the evaluation of the activity of oxygenated oils from plants growing in a specific local environment, largely isolated from external influences.

The work presented is methodically well prepared and the analysis presented in a multidirectional manner allows the evaluation of the material studied. The cell culture studies are well done and the researchers do not look for effects where there are none. I believe that the oils studied can be successfully applied in natural cosmetics with protective and regenerative effects.

Revisions according to reviewer 1 have been highlighted in turquoise in the revised manuscript. We wish to thank the reviewer for support, time and opportunity for sharing these data to the scientific community.

I would kindly to ask you to supplement the information on the origin of the cell lines - the collection from which they were purchased and information on the software in which the calculations of the presented results were performed.

Source of cell lines and software used have been highlighted in the subsection 4.6.1. in the section materials and methods.

Reviewer 2 Report

All scientific names of plants (Cinnamomum camphora, Curcuma  longa, Citrus aurantium, Morinda citrifolia, Petroselinum crispum, Plectranthus amboinicus,  Tosporum senacia, Syzygium coriaceum, and Syzygium samarangense) and microorganisms in full or abbreviated must be in italic

Abstract

 While only S. coriaceum, P. amboinicus (MIC: 0.50 mg/mL) and M. citrifolia (MIC: 2 mg/mL) EOs showed activity against Cutibacterium  acnes. In the same time, all EOs except S. samarangense EO demonstrated activity against Mycobacterium smegmatis  (MIC: 0.125-0.50 mg/mL).

Introduction

Essential oils (EOs) which are aromatic and …….., and antiprotozoal properties [1]. They have been used for cen- Essential oils is repeated many times in introduction try to change this word

Results and Discussion

·         All scientific names of plants below table 1 must be in italic

·         In the current investigation, only P. amboinicus, M. citrifolia and S. coriaceum EOs 142 showed anti-acne activity. P. amboinicus and S. coriaceum EOs showed higher 143 potential (MIC: 0.50 mg/mL) than M. citrifolia EO (2 mg/mL) (Table 1).

·         P. senia EO showed the weakest antifungal effect on C. tropicalis 188 (MIC= 32 mg/mL) (Table 2). Interestingly

·         Remarkably, the same major components, octanoic (38.7%) and hexanoic (20.0%) acids, were revealed in the EO, as was previously reported for M. citrifolia EO investigated in the present study [11], although they varied in their percentages (octanoic acid (78.9%); hexanoic  acid (11.3%)). Correct this sentence I believe: Remarkably, the same major components, octanoic (38.7%) and hexanoic (20.0%) acids, were revealed in the EO, as was previously reported by Jugreet and Mahomoodally (2020) for M. citrifolia EO in their present study[11], although they varied in their percentages (octanoic acid (78.9%); hexanoic acid (11.3%)).

·         All scientific names of plants below table 3 and Table 4. must be in italic

·         M. citrifolia EO exerted no inhibitory effect on any of the cell lines at the highest concentration tested (400 μg/mL) (Table 5). Correct this sentence I believe do not exerted inhibitory on any of the cell lines at the highest tested concentration (400 μg/mL) (Table 5).

·         Besides, the cytotoxic properties of EOs result ……… the effects of another compound [54]. Add more reference(s) to these ideas and discussion, I see that there is only one too all this long paragraph

·         Furthermore, M. citrifolia EO was re ported to be cytotoxic to human colorectal carcinoma (HCT-116) and human breast carci

Materials and Methods

I see that the experimental protocol of Antimicrobial assay, Antiaging assay, Molecular docking and Cell culture are very detailed try to reduce them.

It is better to indicate the origin (company) of used antibiotic (ciprofloxacin and tetracycline, Nystatin, Amphotericin B), Actinomycin D.

Author Response

Revisions according to reviewer 2 have been highlighted in yellow in the revised manuscript.

We wish to thank the reviewer for support, time and opportunity for sharing these data to the scientific community.

  1. All scientific names of plants (Cinnamomum camphora, Curcuma longa, Citrus aurantium, Morinda citrifolia, Petroselinum crispum, Plectranthus amboinicus, Pittosporum senacia, Syzygium coriaceum, and Syzygium samarangense) and microorganisms in full or abbreviated must be in italic.

All scientific names of plant species and microorganisms have been checked and written in italic.

Abstract

  1. While only  coriaceumP. amboinicus(MIC: 0.50 mg/mL) and M. citrifolia (MIC: 2 mg/mL) EOs showed activity against Cutibacterium acnes. In the same time, all EOs except S. samarangense EO demonstrated activity against Mycobacterium smegmatis (MIC: 0.125-0.50 mg/mL).

Has been corrected.

Introduction

  1. Essential oils (EOs) which are aromatic and ……., and antiprotozoal properties [1]. Theyhave been used for cen- Essential oils is repeated many times in introduction try to change this word

This has been amended accordingly.

Results and Discussion

  1. All scientific names of plants below table 1 must be in italic

All scientific names have been put in italic.

  1. In the current investigation, only  amboinicusM. citrifolia and S. coriaceum EOs 142 showed anti-acne activity. P. amboinicus and S. coriaceum EOs showed higher 143 potential (MIC: 0.50 mg/mL) than M. citrifolia EO (2 mg/mL) (Table 1).

Has been corrected.

  1. senia EO showed the weakest antifungal effect on C. tropicalis 188 (MIC= 32 mg/mL) (Table 2). Interestingly

Has been corrected.

  1. Remarkably, the same major components, octanoic (38.7%) and hexanoic (20.0%) acids, were revealed in the EO, as was previously reported for  citrifolia EO investigated in the present study [11], although they varied in their percentages (octanoic acid (78.9%); hexanoic acid (11.3%)). Correct this sentence I believe: Remarkably, the same major components, octanoic (38.7%) and hexanoic (20.0%) acids, were revealed in the EO, as was previously reported by Jugreet and Mahomoodally (2020) for M. citrifolia EO in their present study [11], although they varied in their percentages (octanoic acid (78.9%); hexanoic acid (11.3%)).

The sentence has been corrected.

  1. All scientific names of plants below table 3 and Table 4. must be in italic.

Corrections have been done.

  1. citrifolia EO exerted no inhibitory effect on any of the cell lines at the highest concentration tested (400 μg/mL) (Table 5). Correct this sentence I believe do not exerted inhibitory on any of the cell lines at the highest tested concentration (400 μg/mL) (Table 5).

Sentence has been corrected.

  1. Besides, the cytotoxic properties of EOs result ……… the effects of another compound [54]. Add more reference(s) to these ideas and discussion, I see that there is only one too all this long paragraph

More references have been added to the paragraph (references 54-60).

  1. Furthermore, citrifolia EO was reported to be cytotoxic to human colorectal carcinoma (HCT-116) and human breast carcinoma.

Has been corrected.

Materials and Methods

  1. I see that the experimental protocol of antimicrobial assay, Antiaging assay, Molecular docking and Cell culture are very detailed try to reduce them.

The experimental procedures for the antimicrobial assay, Antiaging assay, Molecular docking and Cell culture have been reduced in the revised manuscript.

  1. It is better to indicate the origin (company) of used antibiotic (ciprofloxacin and tetracycline, Nystatin, Amphotericin B), Actinomycin D.

The companies where the standard drugs have been purchased have been added in the section Material and methods, in the respective subsections.

Reviewer 3 Report

The report is interesting, it discusses important points regarding the investigation of new antimicrobial or anticancer agents, however it mentions some well-known outcomes. The manuscript adds little new information to the group's previous studies (refs 43, 53, 59, 60, 62 and 63). In general, I have nothing against the manuscript, the Material and Methods section is managed very well (except one error mentioned below) and reflects the previous works of the authors, as well as the Results and Discussions section are written clearly and satisfactorily, but the manuscript does not provide any novelties. If there are any novelties, I don't see them, and the authors should highlight them in the next publication.  In addition, there are numerous reports in the scientific literature investigating the same issue (antimicrobial, antifungal, anticancer, antiacne and etc..effects of EOs). I have some comments as follow that should be taken into account:

-there are some errors (typos) in the article,

-in the abstract and somewhere in the article all Latin names must be italic

- the authors did not mention the origin of the tested strains of Mycobacterium and Candida I section Material and methods (the origin of these strains is mentioned only in the tables in the Results section that it is ATTC strains)

Overall, the manuscript brings reduced clinical impact. It only presents in vitro data, only against ATCC strains.  I consider mandatory that clinical strains should be evaluated. In addition, multidrug-resistant strains could have also been employed.

Author Response

The report is interesting, it discusses important points regarding the investigation of new antimicrobial or anticancer agents, however it mentions some well-known outcomes. The manuscript adds little new information to the group's previous studies (refs 43, 53, 59, 60, 62 and 63). In general, I have nothing against the manuscript, the Material and Methods section is managed very well (except one error mentioned below) and reflects the previous works of the authors, as well as the Results and Discussions section are written clearly and satisfactorily, but the manuscript does not provide any novelties. If there are any novelties, I don't see them, and the authors should highlight them in the next publication.  In addition, there are numerous reports in the scientific literature investigating the same issue (antimicrobial, antifungal, anticancer, antiacne and etc..effects of EOs). I have some comments as follow that should be taken into account:

We wish to thank the reviewer for support, time and opportunity for sharing these data to the scientific community.

Revisions according to Reviewer 3 are highlighted in green in the manuscript:

  1. there are some errors (typos) in the article,

Typo errors have been checked throughout the manuscript and have been corrected.

  1. in the abstract and somewhere in the article all Latin names must be italic

All Latin names in the manuscript have been checked and written in italic.

  1. the authors did not mention the origin of the tested strains of Mycobacterium and Candida in section Material and methods (the origin of these strains is mentioned only in the tables in the Results section that it is ATTC strains)

The ATCC strains of Mycobacterium and Candida species used have been added in the section Materials and methods as well.

  1. Overall, the manuscript brings reduced clinical impact. It only presents in vitrodata, only against ATCC strains.  I consider mandatory that clinical strains should be evaluated. In addition, multidrug-resistant strains could have also been employed.

The effects of these essential oils on some clinical and multidrug-resistant bacterial strains have already been studied and published previously (Jugreet, B.S. and Mahomoodally, M.F., 2020. Essential oils from 9 exotic and endemic medicinal plants from Mauritius shows in vitro antibacterial and antibiotic potentiating activities. South African Journal of Botany, 132, pp.355-362).

The aim of conducting more antibacterial studies on these essential oils is to explore if they have wide antimicrobial spectrum of activities.

Round 2

Reviewer 3 Report

Dear authors,

I see that you have edited and corrected the article, but I still don't see anything new in the article. I have changed my opinion on some points, but it still does not change my view of the article. However, I leave the decision on acceptance to the editor, since the article is well written from a methodological point of view and from the point of view of the overall structure of the article.